# Superset Technique for Approximate Recovery in One-Bit Compressed Sensing

**Larkin Flodin**
University of Massachusetts Amherst
Amherst, MA 01003
lflodin@cs.umass.edu

**Venkata Gandikota**
University of Massachusetts Amherst
Amherst, MA 01003
gandikota.venkata@gmail.com

**Arya Mazumdar**
University of Massachusetts Amherst
Amherst, MA 01003
arya@cs.umass.edu

## Abstract

One-bit compressed sensing (1bCS) is a method of signal acquisition under extreme measurement quantization that gives important insights on the limits of signal compression and analog-to-digital conversion. The setting is also equivalent to the problem of learning a sparse hyperplane-classifier. In this paper, we propose a generic approach for signal recovery in nonadaptive 1bCS that leads to improved sample complexity for approximate recovery for a variety of signal models, including nonnegative signals and binary signals. We construct 1bCS matrices that are universal - i.e. work for all signals under a model - and at the same time recover very general random sparse signals with high probability. In our approach, we divide the set of samples (measurements) into two parts, and use the first part to recover the superset of the support of a sparse vector. The second set of measurements is then used to approximate the signal within the superset. While support recovery in 1bCS is well-studied, recovery of superset of the support requires fewer samples, which then leads to an overall reduction in sample complexity for approximate recovery.

## 1 Introduction

Sparsity is a natural property of many real-world signals. For example, image and speech signals are sparse in the Fourier basis, which led to the theory of compressed sensing, and more broadly, sampling theory [12, 7]. In some important multivariate optimization problems with many optimal points, sparsity of the solution is also a measure of 'simplicity' and insisting on sparsity is a common method of *regularization* [19]. While recovering sparse vectors from linear measurements is a well-studied topic, technological advances and increasing data size raises new questions. These include quantized and nonlinear signal acquisition models, such as 1-bit compressed sensing [4]. In 1-bit compressed sensing, linear measurements of a sparse vector are quantized to only 1 bit, e.g. indicating whether the measurement outcome is positive or not, and the task is to recover the vector up to a prescribed Euclidean error with minimum number of measurements. Like compressed sensing, the overwhelming majority of the literature, including this paper, focuses on the nonadaptive setting for the problem.

One of the ways to approximately recover a sparse vector from 1-bit measurements is to use a subset of all the measurements to identify the support of the vector. Next, the remainder of the measurements can be used to approximate the vector within the support. Note that this second set of measurements is also predefined, and therefore the entire scheme is still nonadaptive. Such a method appears in the

context of 'universal' matrix designs in [9, 1]. The resulting schemes are the best known, in some sense, but still result in a large gap between the upper and lower bounds for approximate recovery of vectors.

In this paper we take steps to close these gaps, by presenting a simple yet powerful idea. Instead of using a subset of the measurements to recover the support of the vector exactly, we propose using a (smaller) set of measurements to recover a superset of the support. The remainder of the measurements can then be used to better approximate the vector within the superset. It turns out this idea which we call the "superset technique" leads to optimal number of measurements for universal schemes for several important classes of sparse vectors (for example, nonnegative vectors). We also present theoretical results providing a characterization of matrices that would yield universal schemes for all sparse vectors.

**Prior Results.** While the compressed sensing framework was introduced in [7], it was not until [4] that 1-bit quantization of the measurements was considered as well, to try and combat the fact that taking real-valued measurements to arbitrary precision may not be practical in applications. Initially, the focus was primarily on approximately reconstructing the direction of the signal $\mathbf{x}$ (the quantization does not preserve any information about the magnitude of the signal, so all we can hope to reconstruct is the direction). However, in [10] the problem of support recovery, as opposed to approximate vector reconstruction, was first considered and it was shown that $\mathcal{O}\left(k \log n\right)$ measurements is sufficient to recover the support of a $k$-sparse signal in $\mathbb{R}^n$ with high probability. This was subsequently shown to be tight with the lower bound proven in [3].

All the above results assume that a new measurement matrix is constructed for each sparse signal, and success is defined as either approximately recovering the signal up to error $\epsilon$ in the $\ell_2$ norm (for the approximate vector recovery problem), or exactly recovering the support of the signal (for the support recovery problem), with high probability. Generating a new matrix for each instance is not practical in all applications, which has led to interest in the "universal" versions of the above two problems, where a single matrix must work for support recovery or approximate recovery of all $k$-sparse real signals, with high probability.

Plan and Vershynin showed in [15] that both $\mathcal{O}\left(\frac{k}{\epsilon^6} \log \frac{n}{k}\right)$ and $\mathcal{O}\left(\frac{k}{\epsilon^5} \log^2 \frac{n}{k}\right)$ measurements suffice for universal approximate recovery. The dependence on $\epsilon$ was then improved significantly to $\mathcal{O}\left(k^3 \log \frac{n}{k} + \frac{k}{\epsilon}\right)$ in [9], who also considered the problem of universal support recovery, and showed that for that problem, $\mathcal{O}\left(k^3 \log n\right)$ measurements is sufficient. They showed as well that if we restrict the entries of the signal to be nonnegative (which is the case for many real-world signals such as images), then $\mathcal{O}\left(k^2 \log n\right)$ is sufficient for universal support recovery. The constructions of their measurement matrices are based primarily on combinatorial objects, specifically expanders and Union Free Families (UFFs).

Most recently, [1] showed that a modified version of the UFFs used in [9] called "Robust UFFs" (RUFFs) can be used to improve the upper bound on universal support recovery to $\mathcal{O}\left(k^2 \log n\right)$ for all real-valued signals, matching the previous upper bound for nonnegative signals, and showed this is nearly tight with a lower bound of $\Omega(k^2 \log n / \log k)$ for real signals. They also show that $\mathcal{O}\left(k^2 \log n + \frac{k}{\epsilon}\right)$ measurements suffices for universal approximate recovery.

In tandem with the development of these theoretical results providing necessary and sufficient numbers of measurements for support recovery and approximate vector recovery, there has been a significant body of work in other directions on 1-bit compressed sensing, such as heuristic algorithms that perform well empirically, and tradeoffs between different parameters. More specifically, [11] introduced a gradient-descent based algorithm called Binary Iterative Hard Thresholding (BIHT) which performs very well in practice; later, [13] gave another heuristic algorithm which performs comparably well or better, and aims to allow for very efficient decoding after the measurements are taken. Other papers such as [18] have studied the tradeoff between the amount of quantization of the signal, and the necessary number of measurements.

**Our Results.** We focus primarily on upper bounds in the universal setting, aiming to give constructions that work with high probability for all sparse vectors. In [1], 3 major open questions are given regarding Universal 1-bit Compressed Sensing, which, paraphrasing, are as follows:

1. How many measurements are necessary and sufficient for a matrix to be used to exactly recover all $k$-sparse binary vectors?

Table 1: Upper and lower bounds for 1bCS problems with $k$-sparse signals

| Problem | UB | Explicit UB | LB |
|---|---|---|---|
| Universal Support Recovery ($\mathbf{x} \in \mathbb{R}^n$) | $\mathcal{O}\left(k^2 \log n\right)$ [1] | $\mathcal{O}\left(k^2 \log n\right)^*$ | $\Omega(k^2 \log n / \log k)$ [1] |
| Universal $\epsilon$-approximate Recovery ($\mathbf{x} \in \mathbb{R}^n$) | $\tilde{\mathcal{O}}(\min(k^2 \log \frac{n}{k} + \frac{k}{\epsilon}, \frac{k}{\epsilon} \log \frac{n}{k}))$ [1], [11] | – | $\Omega(k \log \frac{n}{k} + \frac{k}{\epsilon})$ [1] |
| Universal $\epsilon$-approximate Recovery ($\mathbf{x} \in \mathbb{R}^n_{\geq 0}$) | $\mathcal{O}\left(k \log(\frac{n}{k}) + \frac{k}{\epsilon}\right)^*$ | – | $\Omega(k \log \frac{n}{k})$ |
| Universal Exact Recovery ($\mathbf{x} \in \{0, 1\}^n$) | $\mathcal{O}\left(k \log(\frac{n}{k}) + k^{3/2}\right)^*$ | – | $\Omega(k \log \frac{n}{k})$ |
| Non-Universal Support Recovery ($\mathbf{x} \in \mathbb{R}^n$) | $\mathcal{O}\left(k \log n\right)$ [3] | $\mathcal{O}\left(k \log^2 n\right)^*$ | $\Omega(k \log \frac{n}{k})$ [3] |

*Bound proved in this work.

2. What is the correct complexity (in terms of number of measurements) of universal $\epsilon$-approximate vector recovery for real signals?

3. Can we obtain explicit (i.e. requiring time polynomial in $n$ and $k$) constructions of the Robust UFFs used for universal support recovery (yielding measurement matrices with $\mathcal{O}\left(k^2 \log n\right)$ rows)?

In this work we make progress towards solutions to all three Open Questions. Our primary contribution is the "superset technique" which relies on ideas from the closely related sparse recovery problem of group testing [8]; in particular, we show in Theorem 6 that for a large class of signals including all nonnegative (and thus all binary) signals, we can improve the upper bound for approximate recovery by first recovering an $\mathcal{O}\left(k\right)$-sized superset of the support rather than the exact support, then subsequently using Gaussian measurements. The previous best upper bound for binary signals from [11] was $\mathcal{O}\left(k^{3/2} \log n\right)$, which we improve to $\mathcal{O}\left(k^{3/2} + k \log \frac{n}{k}\right)$, and for nonnegative signals was $\mathcal{O}\left(\min(k^2 \log \frac{n}{k} + \frac{k}{\epsilon}, \frac{k}{\epsilon} \log n)\right)$, which we improve to $\mathcal{O}\left(k \log \frac{n}{k} + \frac{k}{\epsilon}\right)$.

Regarding Open Question 3, using results of Porat and Rothschild regarding weakly explicit constructions of Error-Correcting Codes (ECCs) on the Gilbert-Varshamov bound [16], we give a construction of Robust UFFs yielding measurement matrices for support recovery with $\mathcal{O}\left(k^2 \log n\right)$ rows in time that is polynomial in $n$ (though not in $k$) in Theorem 12. Based on a similar idea, we also give a weakly explicit construction for non-universal approximate recovery using only sightly more measurements than is optimal ($\mathcal{O}\left(k \log^2 n\right)$ as opposed to $\mathcal{O}\left(k \log \frac{n}{k}\right)$) in Section 4.2; to our knowledge, explicit constructions in the non-universal setting have not been studied previously. Furthermore, this result gives a single measurement matrix which works for almost all vectors, as opposed to typical non-universal results which work with high probability for a particular vector and matrix pair.

In Appendix C, we give a sufficient condition generalizing the notion of RUFFs for a matrix to be used for universal recovery of a superset of the support for all real signals; while we do not provide constructions, this seems to be a promising direction for resolving Open Question 2.

The best known upper and lower bounds for the various compressed sensing problems considered in this work are presented in Table 1.

## 2 Definitions

We write $M_i$ for the $i$th row of the matrix $M$, and $M_{i,j}$ for the entry of $M$ in the $i$th row and $j$th column. We write vectors $\mathbf{x}$ in boldface, and write $\mathbf{x}_i$ for the $i$th component of the vector $\mathbf{x}$. The set $\{1, 2, \ldots, n\}$ will be denoted by $[n]$, and for any set $S$ we write $\mathcal{P}(S)$ for the power set of $S$ (i.e. the set of all subsets of $S$).

We will write $\text{supp}(\mathbf{x}) \subseteq [n]$ to mean the set of indices of nonzero components of $\mathbf{x}$ (so $\text{supp}(\mathbf{x}) = \{i : \mathbf{x}_i \neq 0\}$), and $||\mathbf{x}||_0$ to denote $|\text{supp}(\mathbf{x})|$.

For a real number $y$, $\text{sign}(y)$ returns 1 if $y$ is strictly positive, $-1$ if y is strictly negative, and 0 if $y = 0$. While this technically returns more than one bit of information, if we had instead defined $\text{sign}(y)$ to be 1 when $y \geq 0$ and $-1$ otherwise, we could still determine whether $y = 0$ by looking at $\text{sign}(y), \text{sign}(-y)$, so this affects the numbers of measurements by only a constant factor. We will not concern ourselves with the constants involved in any of our results, so we have chosen to instead use the more convenient definition.

We will sometimes refer to constructions from the similar "group testing" problem in our results. To this end, we will use the symbol "$\odot$" to represent the group testing measurement between a measurement vector and a signal vector. Specifically, for a measurement $\mathbf{m}$ of length $n$ and signal $\mathbf{x}$

of length $n$, $\mathbf{m} \odot \mathbf{x}$ is equal to 1 if $\operatorname{supp}(\mathbf{m}) \cap \operatorname{supp}(\mathbf{x})$ is nonempty, and 0 otherwise. We will also make use of the "list-disjunct" matrices used in some group testing constructions.

**Definition 1.** *An $m \times n$ binary matrix $M$ is $(k, l)$-list disjunct if for any two disjoint sets $S, T \subseteq \operatorname{col}(M)$ with $|S| = k, |T| = l$, there exists a row in $M$ in which some column from $T$ has a nonzero entry, but every column from $S$ has a zero.*

The primary use of such matrices is that in the group testing model, they can be used to recover a superset of size at most $k + l$ of the support of any $k$-sparse signal $\mathbf{x}$ from applying a simple decoding to the measurement results $M \odot \mathbf{x}$.

In the following definitions, we write $S$ for a generic set that is the domain of the signal. In this paper we consider signals with domain $\mathbb{R}, \mathbb{R}_{\geq 0}$ (nonnegative reals), and $\{0, 1\}$.

**Definition 2.** *An $m \times n$ measurement matrix $M$ can be used for **Universal Support Recovery** of $k$-sparse $\mathbf{x} \in S^n$ (in $m$ measurements) if there exists a decoding function $f : \{-1, 0, 1\}^m \to \mathcal{P}([n])$ such that $f(\operatorname{sign}(M\mathbf{x})) = \operatorname{supp}(\mathbf{x})$ for all $\mathbf{x}$ satisfying $||\mathbf{x}||_0 \leq k$.*

**Definition 3.** *An $m \times n$ measurement matrix $M$ can be used for **Universal $\epsilon$-Approximate Recovery** of $k$-sparse $\mathbf{x} \in S^n$ (in $m$ measurements) if there exists a decoding function $f : \{-1, 0, 1\}^m \to S^n$ such that*

$$\left|\left| \frac{\mathbf{x}}{||\mathbf{x}||_2} - \frac{f(\operatorname{sign}(M\mathbf{x}))}{||f(\operatorname{sign}(M\mathbf{x}))||_2} \right|\right|_2 \leq \epsilon,$$

*for all $\mathbf{x}$ with $||\mathbf{x}||_0 \leq k$.*

## 3 Upper Bounds for Universal Approximate Recovery

Here we present our main result, an upper bound on the number of measurements needed to perform universal $\epsilon$-approximate recovery for a large class of real vectors that includes all binary vectors and all nonnegative vectors. The general technique will be to first use what are known as "list-disjunct" matrices from the group testing literature to recover a superset of the support of the signal, then use Gaussian measurements to approximate the signal within the superset. Because the measurements in the second part are Gaussian, we can perform the recovery within the (initially unknown) superset nonadaptively. When restricting to the class of binary or nonnegative signals, our upper bound improves on existing results and is close to known lower bounds.

First, we need a lemma stating the necessary and sufficient conditions on a signal vector $\mathbf{x}$ in order to be able to reconstruct the results of a single group testing measurement $\mathbf{m} \odot \mathbf{x}$ using sign measurements. To concisely state the condition, we introduce some notation: for a subset $S \subseteq [n]$ and vector $\mathbf{x}$ of length $n$, we write $\mathbf{x}|_S$ to mean the restriction of $\mathbf{x}$ to the indices of $S$.

**Lemma 1.** *Let $\mathbf{m} \in \{0, 1\}^n$ and $\mathbf{x} \in \mathbb{R}^n$. Define $S = \operatorname{supp}(\mathbf{m}) \cap \operatorname{supp}(\mathbf{x})$. If either $S$ is empty or $S$ is nonempty and $\mathbf{m}^T|_S \mathbf{x}|_S \neq 0$, we can reconstruct the result of the group testing measurement $\mathbf{m} \odot \mathbf{x}$ from the sign measurement $\operatorname{sign}(\mathbf{m}^T\mathbf{x})$.*

*Proof.* We observe $\operatorname{sign}(\mathbf{m}^T\mathbf{x})$ and based on that must determine the value of $\mathbf{m} \odot \mathbf{x}$, or equivalently whether $S$ is empty or nonempty. If $\operatorname{sign}(\mathbf{m}^T\mathbf{x}) \neq 0$ then $\mathbf{m}^T\mathbf{x} \neq 0$, so $S$ is nonempty and $\mathbf{m} \odot \mathbf{x} = 1$. Otherwise we have $\operatorname{sign}(\mathbf{m}^T\mathbf{x}) = 0$, in which case we must have $\mathbf{m}^T\mathbf{x} = 0$. If $S$ were nonempty then we would have $\mathbf{m}^T|_S \mathbf{x}|_S = 0$, contradicting our assumption. Therefore in this case we must have $S$ empty and $\mathbf{m} \odot \mathbf{x} = 0$, so for $\mathbf{x}$ satisfying the above condition we can reconstruct the results of a group testing measurement. $\qquad\qquad\square$

For convenience, we use the following property to mean that a signal $\mathbf{x}$ has the necessary property from Lemma 1 with respect to every row of a matrix $M$.

**Property 1.** *Let $M$ be an $m \times n$ matrix, and $\mathbf{x}$ a signal of length $n$. Define $S_i = \operatorname{supp}(M_i) \cap \operatorname{supp}(\mathbf{x})$. Then for every row $M_i$ of $M$, either $S_i$ is empty, or $M_i^T|_{S_i} \mathbf{x}|_{S_i} \neq 0$.*

**Corollary 2.** *Let $M$ be a $(k, l)$-list disjunct matrix, and $\mathbf{x} \in \mathbb{R}^n$ be a $k$-sparse real signal. If Property 1 holds for $M$ and $\mathbf{x}$, then we can use the measurement matrix $M$ to recover a superset of size at most $k + l$ of the support of $\mathbf{x}$ using sign measurements.*

Combining this corollary with results of [6], there exist matrices with $\mathcal{O}\left(k \log(\frac{n}{k})\right)$ rows which we can use to recover an $\mathcal{O}(k)$-sized superset of the support of $\mathbf{x}$ using sign measurements, provided $\mathbf{x}$

satisfies the above condition. Strongly explicit constructions of these matrices exist also, although requiring $\mathcal{O}\left(k^{1+o(1)}\log n\right)$ rows [5].

The other result we need is one that tells us how many Gaussian measurements are necessary to approximately recover a real signal using maximum likelihood decoding. Similar results have appeared elsewhere, such as [11], but we include the proof for completeness.

**Lemma 3.** *There exists a measurement matrix $A$ for 1-bit compressed sensing such that for every pair of $k$-sparse $\mathbf{x}, \mathbf{y} \in \mathbb{R}^n$ with $||\mathbf{x}||_2 = ||\mathbf{y}||_2 = 1$, $\mathrm{sign}(A\mathbf{x}) \neq \mathrm{sign}(A\mathbf{y})$ whenever $||\mathbf{x} - \mathbf{y}||_2 > \epsilon$, provided that*

$$m = \mathcal{O}\left(\frac{k}{\epsilon}\log\left(\frac{n}{k}\right)\right).$$

We will make use of the following facts in the proof.

**Fact 4.** *For all $x \in \mathbb{R}$, $1 - x < e^{-x}$.*

**Fact 5.** *For all $x \in [0, 1]$, $\cos^{-1}(x) \geq \sqrt{2(1-x)}$.*

*Proof of Lemma 3.* Let $A \sim \mathcal{N}^{m \times n}(0, 1)$. For a measurement to separate $\mathbf{x}$ and $\mathbf{y}$, it is necessary that the hyperplane corresponding to some row $\mathbf{a}$ of $A$ lies between $\mathbf{x}$ and $\mathbf{y}$. Thus our goal here is to show that if we take $m$ to be large enough, that all pairs of points at distance $> \epsilon$ will be separated with high probability. Since the rows of $A$ are chosen independently and have Gaussian entries, they are spherically symmetric, and thus the probability that the random hyperplane $\mathbf{a}$ lies between $\mathbf{x}$ and $\mathbf{y}$ is proportional to the angle between them. Let $||\mathbf{x} - \mathbf{y}||_2 > \epsilon$, then we start out by upper bounding the probability that no measurement separates a particular pair $\mathbf{x}$ and $\mathbf{y}$.

Before beginning, recall that for unit vectors $1 - \mathbf{x}^T\mathbf{y} = ||\mathbf{x} - \mathbf{y}||_2^2/2$, so given that $||\mathbf{x} - \mathbf{y}||_2 > \epsilon$, we have $\mathbf{x}^T\mathbf{y} < 1 - \epsilon^2/2$.

$$
\begin{aligned}
\Pr[\mathrm{sign}(\mathbf{ax}) = \mathrm{sign}(\mathbf{ay})] = \quad & 1 - \frac{\cos^{-1}(\mathbf{x}^T\mathbf{y})}{\pi} \\
< \quad & 1 - \frac{\cos^{-1}(1-\epsilon^2/2)}{\pi} \\
\leq \quad & \exp(-\frac{\cos^{-1}(1-\epsilon^2/2)}{\pi}) \quad \text{(from Fact 4).} \\
\leq \quad & \exp(-\frac{\epsilon}{\pi}) \quad\quad \text{(from Fact 5).}
\end{aligned}
$$

As there are $m$ independent measurements, the probability that $\mathbf{x}$ and $\mathbf{y}$ are not separated by any of the $m$ measurements is at most

$$\exp\left(-\frac{m\epsilon}{\pi}\right),$$

so union bounding over all $\binom{n}{k}^2$ pairs of $k$-sparse $\mathbf{x}$ and $\mathbf{y}$, the total probability of error is strictly less than

$$\binom{n}{k}^2 \exp\left(-\frac{m\epsilon}{\pi}\right).$$

This probability becomes less than 1 for $m \geq \frac{\pi}{\epsilon}(2k)\log\frac{n}{k}$, so with this number of measurements there exists a matrix that can perform $\epsilon$-approximate recovery for all pairs of sparse vectors. $\qquad\square$

Note that in the case that we already have a superset of the support of size $\mathcal{O}(k)$, the previous result tells us there exists a matrix with $\mathcal{O}\left(\frac{k}{\epsilon}\log(\frac{\mathcal{O}(k)}{k})\right) = \mathcal{O}\left(\frac{k}{\epsilon}\right)$ rows which can be used to perform $\epsilon$-approximate recovery within the superset. We can do this even nonadaptively, because the rows of the matrix for approximate recovery are Gaussian. Combining this with Corollary 2 and the group testing constructions of [6], we have the following theorem.

**Theorem 6.** *Let $M = \begin{bmatrix} M^{(1)} \\ M^{(2)} \end{bmatrix}$ where $M^{(1)}$ is a $(k, \mathcal{O}(k))$-list disjunct matrix with $\mathcal{O}\left(k\log\frac{n}{k}\right)$ rows, and $M^{(2)}$ is a matrix with $\mathcal{O}\left(\frac{k}{\epsilon}\right)$ rows that can be used for $\epsilon$-approximate recovery within the superset as in Lemma 3, so $M$ consists of $\mathcal{O}\left(k\log(\frac{n}{k}) + \frac{k}{\epsilon}\right)$ rows. Let $\mathbf{x} \in \mathbb{R}^n$ be a $k$-sparse signal. If Property 1 holds for $M^{(1)}$ and $\mathbf{x}$, then $M$ can be used for $\epsilon$-approximate recovery of $\mathbf{x}$.*

**Remark.** *We note that the class of signal vectors* $\mathbf{x}$ *which satisfy the condition in Theorem 6 is actually quite large, in the sense that there is a natural probability distribution over all sparse signals* $\mathbf{x}$ *for which vectors violating the condition occur with probability 0. The details are laid out in Lemma 14.*

As special cases, we have improved upper bounds for nonnegative and binary signals. For ease of comparison with the other results, we assume the binary signal is rescaled to have unit norm, so has all entries either 0 or equal to $1/\sqrt{||\mathbf{x}||_0}$.

**Corollary 7.** *Let* $M = \begin{bmatrix} M^{(1)} \\ M^{(2)} \end{bmatrix}$ *where* $M^{(1)}$ *is a* $(k, \mathcal{O}(k))$*-list disjunct matrix with* $\mathcal{O}\left(k \log \frac{n}{k}\right)$ *rows, and* $M^{(2)}$ *is a matrix with* $\mathcal{O}\left(\frac{k}{\epsilon}\right)$ *rows that can be used for* $\epsilon$*-approximate recovery within the superset as in Lemma 3, so* $M$ *consists of* $\mathcal{O}\left(k \log(\frac{n}{k}) + \frac{k}{\epsilon}\right)$ *rows. Let* $\mathbf{x} \in \mathbb{R}^n$ *be a* $k$*-sparse signal. If all entries of* $\mathbf{x}$ *are nonnegative, then* $M$ *can be used for* $\epsilon$*-approximate recovery of* $\mathbf{x}$.

*Proof.* In light of Theorem 6, we need only note that as all entries of $M^{(1)}$ and $\mathbf{x}$ are nonnegative, Property 1 is satisfied for $M^{(1)}$ and $\mathbf{x}$. $\qquad\square$

**Corollary 8.** *Let* $M = \begin{bmatrix} M^{(1)} \\ M^{(2)} \end{bmatrix}$ *where* $M^{(1)}$ *is a* $(k, \mathcal{O}(k))$*-list disjunct matrix with* $\mathcal{O}\left(k \log \frac{n}{k}\right)$ *rows, and* $M^{(2)}$ *is a matrix with* $\mathcal{O}\left(k^{3/2}\right)$ *rows that can be used for* $\epsilon$*-approximate recovery (with* $\epsilon < 1/\sqrt{k}$*) within the superset as in Corollary 2 , so* $M$ *consists of* $\mathcal{O}\left(k \log(\frac{n}{k}) + k^{3/2}\right)$ *rows. Let* $\mathbf{x} \in \mathbb{R}^n$ *be the* $k$*-sparse signal vector. If all nonzero entries of* $\mathbf{x}$ *are equal, then* $M$ *can be used for exact recovery of* $\mathbf{x}$.

*Proof.* Here we use the fact that if we perform $\epsilon$-approximate recovery using $\epsilon < 1/\sqrt{k}$ then as the minimum possible distance between any two $k$-sparse rescaled binary vectors is $1/\sqrt{k}$, we will recover the signal vector exactly. $\qquad\square$

## 4 Explicit Constructions

### 4.1 Explicit Robust UFFs from Error-Correcting Codes

In this section we explain how to combine several existing results in order to explicitly construct Robust UFFs that can be used for support recovery of real vectors. This partially answers Open Problem 3 from [1].

**Definition 4.** *A family of sets* $\mathcal{F} = \{B_1, B_2, \ldots, B_n\}$ *with each* $B_i \subseteq [m]$ *is an* $(n, m, d, k, \alpha)$*-Robust-UFF if* $|B_i| = d, \forall i$, *and for every distinct* $j_0, j_1, \ldots, j_k \in [n]$, $|B_{j_0} \cap (B_{j_1} \cup B_{j_2} \cup \cdots \cup B_{j_k})| < \alpha |B_{j_0}|$.

It is shown in [1] that nonexplicit $(n, m, d, k, 1/2)$-Robust UFFs exist with $m = \mathcal{O}\left(k^2 \log n\right), d = \mathcal{O}\left(k \log n\right)$ which can be used to exactly recover the support of any $k$-sparse real vector of length $n$ in $m$ measurements.

The results we will need are the following, where the $q$-ary entropy function $H_q$ is defined as

$$H_q(x) = x \log_q(q-1) - x \log_q x - (1-x) \log_q(1-x).$$

**Theorem 9** ([16] Thm. 2)**.** *Let* $q$ *be a prime power,* $m$ *and* $k$ *positive integers, and* $\delta \in [0, 1]$. *Then if* $k \leq (1 - H_q(\delta))m$, *we can construct a* $q$*-ary linear code with rate* $\frac{k}{m}$ *and relative distance* $\delta$ *in time* $\mathcal{O}\left(mq^k\right)$.

**Theorem 10** ([1] Prop. 17)**.** *Given a* $q$*-ary error correcting code with rate* $r$ *and relative distance* $(1 - \beta)$, *we can construct a* $(q^{rd}, qd, d, 1, \beta)$*-Robust-UFF.*

**Theorem 11** ([1] Prop. 15)**.** *If* $\mathcal{F}$ *is an* $(n, m, d, 1, \alpha/k)$*-Robust-UFF, then* $\mathcal{F}$ *is also an* $(n, m, d, k, \alpha)$*-Robust-UFF.*

By combining the above three results, we have the following.

**Theorem 12.** *We can explicitly construct an $(n, m, d, k, \alpha)$-Robust UFF with $m = \mathcal{O}\left(\frac{k^2 \log n}{\alpha^2}\right)$ and $d = \mathcal{O}\left(\frac{k \log n}{\alpha}\right)$ in time $\mathcal{O}\left((k/\alpha)^k\right)$.*

*Proof.* First, we instantiate Theorem 9 to obtain a $q$-ary code $\mathcal{C}$ of length $d$ with $q = \mathcal{O}(k/\alpha)$, relative distance $\delta = \frac{k-\alpha}{k}$, and rate $r = 1 - H_q(\delta)$ in time $\mathcal{O}(q^k)$.

Applying Theorem 10 to this code results in an $(n, m, d, 1, \beta)$-Robust-UFF $\mathcal{F}$ where $n = q^{rd}$, $m = qd$, $\beta = 1 - \delta$. By Theorem 11, $\mathcal{F}$ is also an $(n, m, d, k, \beta k)$-Robust UFF. Plugging back in the parameters of the original code,

$$
m = qd = \frac{q \log n}{r \log q} = \frac{q \log n}{(1 - H_q((k - \alpha)/k)) \log q} = \mathcal{O}\left(\frac{k^2 \log n}{\alpha^2}\right),
$$

$$
\beta k = (1 - \delta)k = (1 - \frac{k - \alpha}{k})k = k - (k - \alpha) = \alpha.
$$

$\square$

While the time needed for this construction is not polynomial in $k$ (and therefore the construction is not strongly explicit) as asked for in Open Question 3 of [1], this at least demonstrates that there exist codes with sufficiently good parameters to yield Robust UFFs with $m = \mathcal{O}(k^2 \log n)$.

### 4.2 Non-Universal Approximate Recovery

If instead of requiring our measurement matrices to be able to recover all $k$-sparse signals simultaneously (i.e. to be universal), we can instead require only that they are able to recover "most" $k$-sparse signals. Specifically, in this section we will assume that the sparse signal is generated in the following way: first a set of $k$ indices is chosen to be the support of the signal uniformly at random. Then, the signal is chosen to be a uniformly random vector from the unit sphere on those $k$ indices. We relax the requirement that the supports of all $k$-sparse signals can be recovered exactly (by some decoding) to the requirement that we can identify the support of a $k$-sparse signal with probability at least $1 - \delta$, where $\delta \in [0, 1)$. Note that even when $\delta = 0$, this is a weaker condition than universality, as the space of possible $k$-sparse signals is infinite.

It is shown in [3] that a random matrix construction using $\mathcal{O}(k \log n)$ measurements suffices to recover the support with error probability approaching 0 as $k$ and $n$ approach infinity. The following theorem shows that we can explicitly construct a matrix which works in this setting, at the cost of slightly more measurements (about $\mathcal{O}(k \log^2(n))$).

**Theorem 13.** *We can explicitly construct measurement matrices for Support Recovery (of real vectors) with $m = \mathcal{O}\left(k \frac{\log(n)}{\log k} \log(\frac{n}{\delta})\right)$ rows that can exactly determine the support of a $k$-sparse signal with probability at least $1 - \delta$, where the signals are generated by first choosing the size $k$ support uniformly at random, then choosing the signal to be a uniformly random vector on the sphere on those $k$ coordinates.*

To prove this theorem, we need a lemma which explains how we can use sign measurements to "simulate" group testing measurements with high probability. Both the result and proof are similar to Lemma 1, with the main difference being that given the distribution described above, the vectors violating the necessary condition in Lemma 1 occur with zero probability and so can be safely ignored. For this lemma, we do not need the further assumption made in Theorem 13 that the distribution over support sets is uniform. The proof is presented in Appendix A.

**Lemma 14.** *Suppose we have a measurement vector $\mathbf{m} \in \{0, 1\}^n$, and a $k$-sparse signal $\mathbf{x} \in \mathbb{R}^n$. The signal $\mathbf{x}$ is generated randomly by first picking a subset of size $k$ from $[n]$ (using any distribution) to be the support, then taking $\mathbf{x}$ to be a uniformly random vector on the sphere on those $k$ coordinates. Then from $\text{sign}(\mathbf{m}^T \mathbf{x})$, we can determine the value of $\mathbf{m} \odot \mathbf{x}$ with probability 1.*

As the above argument works with probability 1, we can easily extend it to an entire measurement matrix $M$ with any finite number of rows by a union bound, and recover all the group testing measurement results $M \odot \mathbf{x}$ with probability 1 as well. This means we can leverage the following result from [14]:

**Theorem 15** ([14] Thm. 5). *When* $\mathbf{x} \in \{0, 1\}^n$ *is drawn uniformly at random among all $k$-sparse binary vectors, there exists an explicitly constructible group testing matrix $M$ with $m = \mathcal{O}\left(\frac{k}{\log k} \log(n) \log(\frac{n}{\delta})\right)$ rows which can exactly identify $\mathbf{x}$ from observing the measurement results $M \odot \mathbf{x}$ with probability at least $1 - \delta$.*

Combining this with the lemma above, we can use the matrix $M$ from Theorem 15 with $m = \mathcal{O}\left(\frac{k}{\log k} \log n \log(\frac{n}{\delta})\right)$ rows (now representing sign measurements) to exactly determine the support of $\mathbf{x}$ with probability at least $1 - \delta$; we first use Lemma 14 to recover the results of the group testing tests $M \odot \mathbf{x}$ with probability 1, and can then apply the above theorem using the results of the group testing measurements.

We can also use this construction for approximate recovery rather than support recovery using Lemma 3, by appending $\mathcal{O}\left(\frac{k}{\epsilon}\right)$ rows of Gaussian measurements to $M$, first recovering the exact support, then doing approximate recovery within that support. This gives a matrix with about $\mathcal{O}\left(k \log^2(n) + \frac{k}{\epsilon}\right)$ rows for non-universal approximate recovery of real signals, where the top portion is explicit.

**Remark.** *Above, we have shown that in the non-universal setting, we can use constructions from group testing to recover the exact support with high probability, and then subsequently perform approximate recovery within that exact support. If we are interested only in performing approximate recovery, we can apply our superset technique here as well; Lemma 14 implies also that using a $(k, \mathcal{O}(k))$-list disjunct matrix we can with probability 1 recover an $\mathcal{O}(k)$-sized superset of the support, and such matrices exist with $\mathcal{O}\left(k \log \frac{n}{k}\right)$ rows. Following this, we can use $\mathcal{O}\left(\frac{k}{\epsilon}\right)$ more Gaussian measurements to recover the signal within the superset. This gives a non-universal matrix with $\mathcal{O}\left(k \log \frac{n}{k} + \frac{k}{\epsilon}\right)$ rows for approximate recovery, the top part of which can be made strongly explicit with only slightly more measurements ($\mathcal{O}\left(k^{1+o(1)} \log \frac{n}{k}\right)$ vs. $\mathcal{O}\left(k \log \frac{n}{k}\right)$).*

## 5 Experiments

In this section, we present some empirical results relating to the use of our superset technique in approximate vector recovery for real-valued signals. To do so, we compare the average error (in $\ell_2$ norm) of the reconstructed vector from using an "all Gaussian" measurement matrix to first using a small number of measurements to recover a superset of the support of the signal, then using the remainder of the measurements to recover the signal within that superset via Gaussian measurements. We have used the well-known BIHT algorithm of [11] for recovery of the vector both using the all Gaussian matrix and within the superset, but we emphasize that this superset technique is highly general, and could just as easily be applied on top of other decoding algorithms that use only Gaussian measurements, such as the "QCoSaMP" algorithm of [17].

To generate random signals $\mathbf{x}$, we first choose a size $k$ support uniformly at random among the $\binom{n}{k}$ possibilities, then for each coordinate in the chosen support, generate a random value from $\mathcal{N}(0, 1)$. The vector is then rescaled so that $\|\mathbf{x}\|_2 = 1$.

For the dotted lines in Figure 1 labeled "all Gaussian," for each value of $(n, m, k)$ we performed 500 trials in which we generated an $m \times n$ matrix with all entries in $\mathcal{N}(0, 1)$. We then used BIHT (run either until convergence or 1000 iterations, as there is no convergence guarantee) to recover the signal from the measurement matrix and measurement outcomes.

For the solid lines in Figure 1 labeled "$4k \log n$ Superset," we again performed 500 trials for each value of $(n, m, k)$ where in each trial we generated a measurement matrix $M = \begin{bmatrix} M^{(1)} \\ M^{(2)} \end{bmatrix}$ with $m$ rows in total. Each entry of $M^{(1)}$ is a Bernoulli random variable that takes value 1 with probability $\frac{1}{k+1}$ and value 0 with probability $\frac{k}{k+1}$; there is evidence from the group testing literature [3, 2] that this probability is near-optimal in some regimes, and it appears also to perform well in practice; see Appendix B for some empirical evidence. The entries of $M^{(2)}$ are drawn from $\mathcal{N}(0, 1)$. We use a standard group testing decoding (i.e., remove any coordinates that appear in a test with result 0) to determine a superset based on $\mathbf{y}_1 = \text{sign}(M^{(1)}\mathbf{x})$, then use BIHT (again run either until convergence or 1000 iterations) to reconstruct $\mathbf{x}$ within the superset using the measurement results $\mathbf{y}_2 = \text{sign}(M^{(2)}\mathbf{x})$. The number of rows in $M^{(1)}$ is taken to be $m_1 = 4k \log_{10}(n)$ based on the

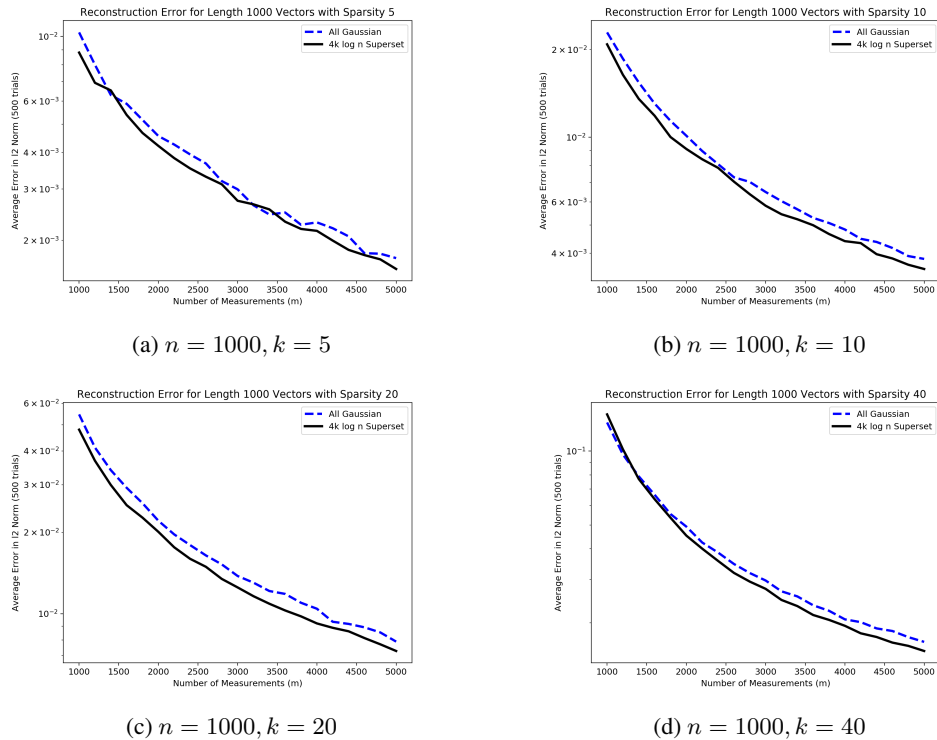

| (a) $n = 1000, k = 5$ | (b) $n = 1000, k = 10$ |
| (c) $n = 1000, k = 20$ | (d) $n = 1000, k = 40$ |

Figure 1: Average error of reconstruction for different sparsity levels with and without use of matrix for superset of support recovery

fact that with high probability $Ck \log n$ rows for some constant $C$ should be sufficient to recover an $\mathcal{O}(k)$-sized superset, and the remainder $m_2 = (m - m_1)$ of the measurements are used in $M^{(2)}$.

We display data only for larger values of $m$, to ensure there are sufficiently many rows in both portions of the measurement matrix. From Figure 1 one can see that in this regime, using a small number of measurements to first recover a superset of the support provides a modest improvement in reconstruction error compared to the alternative. In the higher-error regime when there are simply not enough measurements to obtain an accurate reconstruction, as can be seen in the left side of the graph in Figure 1d, the two methods perform about the same. In the empirical setting, our superset of support recovery technique can be viewed as a very flexible and low overhead method of extending other existing 1bCS algorithms which use only Gaussian measurements, which are quite common.

*Acknowledgements:* This research is supported in part by NSF CCF awards 1618512, 1642658, and 1642550 and the UMass Center for Data Science.

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
