[Supplementary Material]

# Supplementary Material: Superset Technique for Approximate Recovery in One-Bit Compressed Sensing

## A Proof of Lemma 14

**Lemma** (Lemma 14). *Suppose we have a known measurement vector $\mathbf{m} \in \{0,1\}^n$, and an unknown $k$-sparse signal $\mathbf{x} \in \mathbb{R}^n$. The signal $\mathbf{x}$ is generated randomly by first picking a subset of size $k$ from $[n]$ (using any distribution) to be the support, then taking $\mathbf{x}$ to be a uniformly random vector on the sphere on those $k$ coordinates. Then from $sign(\mathbf{m}^T\mathbf{x})$, we can determine the value of $\mathbf{m} \odot \mathbf{x}$ with probability 1.*

*Proof.* We assume without loss of generality that $\mathbf{x}$ is supported on the first $k$ coordinates; the remainder of the argument does not depend specifically on the choice of support, so this is purely for notational convenience. If $sign(\mathbf{m}^T\mathbf{x}) \neq 0$, then immediately we must have $\mathbf{m} \odot \mathbf{x} = 1$, as $\mathbf{m}^T\mathbf{x} \neq 0$.

Otherwise if $sign(\mathbf{m}^T\mathbf{x}) = 0$, we must have $\mathbf{m}^T\mathbf{x} = 0$. This leaves two cases: either $\mathbf{m} \odot \mathbf{x} = 0$, or $\mathbf{x}$ is orthogonal to $\mathbf{m}$ and $\mathbf{m} \odot \mathbf{x} = 1$. In the latter case $\mathbf{x}$ satisfies the equation

$$\sum_{i=1}^k \mathbf{m}_i\mathbf{x}_i = 0 \iff \mathbf{m}_1\mathbf{x}_1 = -\left(\sum_{i=2}^k \mathbf{m}_i\mathbf{x}_i\right).$$

Let $\mathbf{z}$ be a random vector formed by using the same distribution as that used to determine the support of $\mathbf{x}$ in order to determine the support, then within that support drawing $k$ variables $Z_i \sim \mathcal{N}(0,1)$ to be the $k$ coordinates, and finally rescaling so that $||\mathbf{z}||_2 = 1$. It is well-known that the distribution of such $\mathbf{z}$ is identical to the distribution of $\mathbf{x}$, thus the probability that $\mathbf{z}$ is orthogonal to $\mathbf{m}$ is the same as the probability that $\mathbf{x}$ is orthogonal to $\mathbf{m}$. We proceed by showing the probability $\mathbf{z}$ is orthogonal to $\mathbf{m}$ is 0.

If $\mathbf{z}$ is orthogonal to $\mathbf{m}$, then as above we must have

$$\frac{\mathbf{m}_1 Z_1}{||\mathbf{z}||_2} = -\frac{\left(\sum_{i=2}^k \mathbf{m}_i Z_i\right)}{||\mathbf{z}||_2}$$
$$\implies Z_1 = -\left(\sum_{i=2}^k \mathbf{m}_i Z_i\right)/\mathbf{m}_1.$$

Thus in order for $\mathbf{z}$ to lie in the nullspace of $\mathbf{m}$, it is necessary that $Z_1$ takes a specific value determined by the other $k-1$ $Z_i$; as $Z_1$ is drawn independently of the other $Z_i$ and from a continuous distribution, this happens with probability 0. We conclude that the same is true for $\mathbf{x}$, and thus when $sign(\mathbf{m}^T\mathbf{x}) = 0$ we assume that $\mathbf{m} \odot \mathbf{x} = 0$, and are correct with probability 1. $\square$

|  | (a) $n = 1000, k = 10$ | (b) $n = 1000, k = 20$ | (c) $n = 1000, k = 40$ |

Figure 2: Average size of superset following group testing decoding for different sparsity levels as Bernoulli probability of measurement matrix varies. Vertical line highlights $\frac{1}{k+1}$.

## B Empirical Evidence for Experimental Choice of Bernoulli Probability

In this section, we provide some empirical evidence that the choice of $\frac{1}{k+1}$ for the Bernoulli probability of the experiments in Section 5 is reasonable.

Figure 2 shows the average size of the superset using a matrix with Bernoulli entries (i.e. each value is 1 with probability $p$ and 0 otherwise) following a group testing decoding. The different lines represent different numbers of measurements used in the Bernoulli matrix, and different plots show different sparsity levels. All vectors had length 1000, and were constructed randomly by first choosing a size $k$ support set uniformly at random, then drawing a random value from $\mathcal{N}(0, 1)$ for each coordinate in the support set and normalizing so that $||\mathbf{x}||_2 = 1$. 1000 trials were performed for each tuple $(n, k, p)$ of values.

The vertical line overlaid atop the other curves in Figure 2 indicates where the Bernoulli probability is equal to $\frac{1}{k+1}$. For all three sparsity levels, it appears that this value is very close to achieving the minimum size superset for a given number of measurements. Furthermore, the fact that the curves all have relatively wide basins around the minimum indicates that any value close to the minimum should perform fairly well.

## C Sufficient Condition for Universal Support Recovery of Real Vectors

The goal in this section is to give sufficient conditions on a measurement matrix in order to be able to recover a superset of the support of an unknown $k$-sparse signal $\mathbf{x} \in \mathbb{R}^n$ using 1-bit sign measurements, by generalizing the definition of "Robust UFF" given in [1].

In this section we will work primarily with matrix columns rather than rows, so to this end for any matrix $B \in \mathbb{R}^{m \times n}$, here we let $B_j$ denote its $j$-th column. For any sets $X \subseteq [m]$ and $Y \subseteq [n]$, let $B[X : Y]$ denote the submatrix of $B$ restricted to rows indexed by $X$ and columns indexed by $Y$. Let $\mathrm{wt}(\mathbf{x})$ denote the size of the support of $x$, i.e. $\mathrm{wt}(\mathbf{x}) = |\mathrm{supp}(\mathbf{x})|$. We say $\mathbf{x}$ has full support if $\mathrm{wt}(\mathbf{x}) = n$.

In order to recover the superset of the support of $x$ using the sign measurements $\mathrm{sign}(B\mathbf{x}) \in \{-1, 0, 1\}^m$, we use the algorithm of [1] (Algorithm 1). For any subset of $k$ columns $S \subset [n]$, $|S| \leq k$, define $T_S := \{j \in [n] \setminus S : |\mathrm{supp}(B_j) \cap (\cup_{i \in S} \mathrm{supp}(B_i))| \geq \frac{1}{2} \mathrm{wt}(B_j)\}$. These are the columns outside of the subset $S$ that have large intersection with the union of the $k$ columns indexed by $S$.

[1] show that if $B$ is a Robust UFF with sufficient parameters, then their algorithm recovers the exact support of $x$. Algorithm 1 computes the intersection of the support of each column $B_j$ with the output $b := \mathrm{sign}(B\mathbf{x})$. It includes the index $j$ in the estimated support if the intersection is sufficiently large. The property of a Robust UFF ensures that the estimated support is exactly the support of $\mathbf{x}$.

We relax the definition of an $(n, m, d, k, \alpha)$-Robust UFF to allow a few false positives, since we only require a superset of the support of $x$ rather than the exact support. The allowable size of $T_S$ controls the number of false positives. Note that allowing $|T_S| \geq 1$ might induce some false negatives as well, thus to avoid this possibility we need to ensure that no column of $B$ in the support of $\mathbf{x}$ has too many zero test results. In general, zero test results can occur when $\mathbf{x}$ lies in the nullspace of many rows of $B$ that have a nonempty intersection with the support of $\mathbf{x}$. We construct the matrix $B$ to avoid such situations.

Input: $B : (n, m, d, k, 1/2)$-Robust UFF, $\mathbf{x} \in \mathbb{R}^n$, unknown $k$-sparse vector.
Let $\mathbf{b} := \text{sign}(Bx)$.
$\hat{S} = \emptyset$.
for $j \in [n]$,
$\quad$ if $|\text{supp}(B_j) \cap \text{supp}(\mathbf{b})| > \frac{d}{2}$,
$\quad\quad$ $\hat{S} \leftarrow \{j\}$
Return $\hat{S}$.

**Algorithm 1:** Support recovery via Robust-UFF

For any subset $S \subseteq [n]$, and any $j \in T_S$, define $L_{S,j} := \{t \in \text{supp}(B_j) \cap (\cup_{i \in S} \text{supp}(B_i))\} \subseteq [m]$. These are the rows in the support of $B_j$ that intersect with the support of the columns of $B$ indexed by $S$. In order to ensure that the algorithm does not introduce any false negatives, we want the output vector $\mathbf{b}$ to have not many zeros in rows corresponding to $L_{S,j}$. Let us define $A_{S,j} := B[L_{S,j} : S \cup \{j\}]$ to be the matrix restricted to the rows in $L_{S,j}$ and columns of $S \cup \{j\}$. Note that since $j \in T_S$, $|L_{S,j}| \geq \frac{\text{wt}(B_j)}{2}$, therefore $A_{S,j}$ has at least $\frac{\text{wt}(B_j)}{2}$ rows. We now define a list-Robust UFF as follows:

**Definition 5** (List-RUFF). *A real matrix $B \in \mathbb{R}^{m \times n}$ is called an $(m, n, d, k, 1/2, \ell)$-list Robust UFF if $\text{wt}(B_j) = d$ for all $j \in [n]$, and for all subsets $S \subseteq [n]$, $|S| \leq k$, the following properties hold:*

1. $|T_S| < \ell$.

2. *For any $j \in T_S$, and any $\mathbf{x} \in \mathbb{R}^{|S|}$ with full support, $\text{wt}(A_{S,j}\mathbf{x}) > |L_{S,j}| - \frac{1}{2}\text{wt}(B_j)$.*

The first condition ensures that the Algorithm 1 introduces at most $\ell$ false positives. The second condition is used to ensure that no $k$-sparse vector $\mathbf{x}$ is in the nullspace of too many rows of $B$, and therefore Algorithm 1 will not yield any false negatives.

Next we show that Algorithm 1 recovers a superset of size at most $k + \ell$ given a measurement matrix $B$ which is an $(m, n, d, k, 1/2, \ell)$-list RUFF.

**Theorem 16.** *Let $\mathbf{x} \in \mathbb{R}^n$ be an unknown $k$-sparse vector with $\text{supp}(\mathbf{x}) = S^*$. If $B$ is an $(n, m, d, k, 1/2, \ell)$-list RUFF, then Algorithm 1 returns $\hat{S}$ such that $S^* \subseteq \hat{S} \subseteq S^* \cup T_{S^*}$.*

*Proof.* We first show that $\hat{S} \subseteq S^* \cup T_{S^*}$. We in fact prove the contrapositive, i.e. if $j \notin S^* \cup T_{S^*}$, then $j \notin \hat{S}$. Let $j \in [n] \setminus (S^* \cup T_{S^*})$. By definition of $T_{S^*}$, we know that $\text{supp}(B_j)$ does not intersect $\cup_{i \in S^*} \text{supp}(B_i)$ in too many places, i.e. $|\text{supp}(B_j) \cap (\cup_{i \in S^*} \text{supp}(B_i))| < \frac{\text{wt}(B_j)}{2}$. Consider all the rows $t \in \text{supp}(B_j) \setminus (\cup_{i \in S^*} \text{supp}(B_i))$. Note that for all these rows, $b_t = 0$. Therefore,

$$|\text{supp}(b) \cap \text{supp}(B_j)| \leq |\text{supp}(B_j)| - |\text{supp}(B_j) \setminus (\cup_{i \in S^*} \text{supp}(B_i))|$$
$$= |\text{supp}(B_j) \cap (\cup_{i \in S^*} \text{supp}(B_i))| < \frac{\text{wt}(B_j)}{2}.$$

From Algorithm 1, it then follows that $j \notin \hat{S}$.

To show that every $j \in S^*$ is included in $\hat{S}$, we need to show that for every such $j$, $|\text{supp}(b) \cap \text{supp}(B_j)| > \frac{\text{wt}(B_j)}{2}$. This is equivalent to showing that there are not too many zeros in the rows of $b$ corresponding to rows in $\cup_{i \in S^*} \text{supp}(B_i)$. Let $j \in S^*$ be any column in the support of $\mathbf{x}$. Let us partition $\text{supp}(B_j)$ into two groups. Let $S_j^* := S^* \setminus \{j\}$. Define

$$G_1 := \{t \in \text{supp}(B_j) \cap \left(\cup_{i \in S_j^*} \text{supp}(B_i)\right)\}, \text{ and}$$
$$G_2 := \text{supp}(B_j) \setminus G_1 = \{t \in \text{supp}(B_j) \setminus \left(\cup_{i \in S_j^*} \text{supp}(B_i)\right)\}.$$

Note that for all $t \in G_2$, $b_t \neq 0$ since $b_t = \mathbf{x}_j \cdot B_j(t) \neq 0$ since $j \in \text{supp}(\mathbf{x})$. Therefore, $G_2 \subseteq \text{supp}(\mathbf{b}) \cap \text{supp}(B_j)$. We can without loss of generality assume that $j \in T_{S_j^*}$. Otherwise, by definition of $T_{S_j^*}$ it follows that $|G_2| > \frac{\text{wt}(B_j)}{2}$, and Algorithm 1 includes $j \in \hat{S}$.

We now show that $\mathbf{b}_t \neq 0$ for many $t \in G_1$. In particular, we show that $\mathbf{b}_t$ is zero for at most $\frac{\mathrm{wt}(B_j)}{2}$ indices in $G_1$. This follows from the property of the list-RUFF. Consider the following submatrix of $B$, $A_{S_j^*,j} := B[G_1, S^*] = B\left[L_{S_j^*,j} : S_j^* \cup \{j\}\right]$. Since $j \in T_{S_j^*}$, $|G_1| > \mathrm{wt}(B_j)/2$, and therefore $A_{S_j^*,j}$ has at least $\mathrm{wt}(B_j)/2$ rows, and at most $k$ columns.

From the definition of list-RUFF, we know that for any $\mathbf{z} \in \mathbb{R}^{|S^*|}$ with full support, $\mathrm{wt}(A_{S_j^*,j}\,\mathbf{z}) > |L_{S,j}| - \frac{1}{2}\mathrm{wt}(B_j) = |G_1| - \frac{1}{2}\mathrm{wt}(B_j)$. Therefore, for $\mathbf{x}$ that is supported on $S^*$, $\mathbf{b}_t \neq 0$ for at least $|G_1| - \frac{1}{2}\mathrm{wt}(B_j)$ indices in $G_1$.

Combining these observations, it follows that

$$|\mathrm{supp}(\mathbf{b}) \cap \mathrm{supp}(B_j)| > |G_1| - \frac{1}{2}\mathrm{wt}(B_j) + |G_2| = \frac{1}{2}\mathrm{wt}(B_j).$$

Therefore the fact that $j \in \hat{S}$ follows from Algorithm 1.

$\square$

In light of this, a possible direction for improving the current upper bound for universal approximate recovery of real vectors would be to show the existence of $(m, n, d, k, 1/2, \mathcal{O}(k))$-list RUFFs with $m = o(k^2 \log(\frac{n}{k}))$. This would immediately yield a measurement matrix with $\mathcal{O}\left(m + \frac{k}{\epsilon}\right)$ rows that could be used for universal $\epsilon$-approximate recovery. We show below via a simple probabilistic construction that matrices satisfying the first property in definition 5 with $m = O(k \log n)$ and $\ell = O(k)$ exist, but leave open the question of whether $O(k \log n)$ rows suffices also for the second property, or whether $O(k^2 \log n)$ rows are necessary.

**Theorem 17.** *There exist matrices $B \in \mathbb{R}^{m \times n}$ satisfying $\mathrm{wt}(B_j) = \frac{m}{k}$ for all columns $B_j$ and for every subset of columns $S \subseteq [n]$, $|S| \leq k$, we have $|T_S| < \ell$, under the assumptions that $m = \Omega(k \log n)$, $k = o(n/\log(n))$, and $\ell = \Omega(k)$.*

*Proof.* We will construct $B$ by drawing a set $S_j \subseteq [m]$ of size $d = \frac{m}{k}$ uniformly at random among all such sets for each column of $B$. If $i \in S_j$ then we set the $i$th entry of $B_j$ to 1, otherwise 0. Now we must show that with probability less than 1 there does not exist any subset $S$ of at most $k$ columns of $B$ with $|T_S| \geq \ell = \Omega(k)$.

Recall that by definition,

$$T_S = \{j \in [n] \setminus S : |\mathrm{supp}(B_j) \cap (\cup_{i \in S} \mathrm{supp}(B_i))| \geq \frac{1}{2}\mathrm{wt}(B_j)\},$$

or in other words, $T_S$ is the set of "confusable" columns for the subset $S$ of columns of $B$. The event that we wish to avoid is that there exists a set $S$ of $k + \ell$ "bad" columns for which the union of the supports of a subset $S' \subseteq S$ of $k$ of those columns has a large intersection with the supports of all of the remaining $\ell$ columns. Since the columns of $B$ are all chosen independently, we have

$$\Pr[B \text{ has a bad set } S \text{ of } k + \ell \text{ columns}] \tag{1}$$

$$\leq \binom{n}{\ell + k} \Pr[S \subseteq \mathrm{col}(B) \text{ is a bad set of } k + \ell \text{ columns}] \tag{2}$$

$$\leq \binom{n}{\ell + k}\binom{\ell + k}{k} \Pr[\text{ for all } \ell \text{ columns } B_i \text{ in } S \setminus S', i \in T_{S'}] \tag{3}$$

$$\leq \binom{n}{\ell + k}\binom{\ell + k}{k}(\Pr[i \in T_{S'}])^\ell. \tag{4}$$

Now we can assume we have a fixed set $S'$ of $k$ columns and another fixed column $B_i$, and we want to upper bound the probability that more than half the $d = \frac{m}{k}$ nonzero entries of $B_i$ lie in $\cup_{j \in S'} \mathrm{supp}(B_j)$. Let $X_j$ be the binary random variable that is equal to 1 if and only if the $j$th entry of $B_i$ is nonzero and lies in $\cup_{j \in S'} \mathrm{supp}(B_j)$. Since every column has weight exactly $d$, $|\cup_{j \in S'} \mathrm{supp}(B_j)| \leq kd$, thus for any $j$ $\Pr[X_j = 1] \leq \frac{kd}{n}$. Then by linearity of expectation we conclude that

$$E[\sum_{j=1}^{m} X_j] = d\Pr[X_j = 1] \leq \frac{kd^2}{n}. \tag{5}$$

While the $X_j$ are not independent, if some $X_j = 1$ then it is less likely that a different random variable $X_{j'} = 1$ as there are less coordinates remaining in $\cup_{j \in S'} \operatorname{supp}(B_j)$. Since the $X_j$ are negatively correlated we can apply a Chernoff bound:

$$\Pr[\sum_{j=1}^{m} X_j \geq \frac{n}{2m} E[\sum_{j=1}^{m} X_j]] < \left( \frac{e^{(n/2m)-1}}{(n/2m)^{n/2m}} \right)^{m^2/nk} < \left( \frac{2em}{n} \right)^{m/2k}. \tag{6}$$

Note that

$$\frac{n}{2m} E[\sum_{j=1}^{m} X_j] \leq \frac{n}{2m} \cdot \frac{kd^2}{n} = \frac{d}{2}, \tag{7}$$

so in order for the sum of the $X_j$ to exceed $\frac{d}{2}$ (which would mean the corresponding fixed column has large overlap with the union of the set of $k$ columns), it must also exceed $\frac{n}{2m} E[\sum_{j=1}^{m} X_j]$.

Combining everything above,

$$\Pr[B \text{ has a bad set } S \text{ of } k + \ell \text{ columns}] \tag{8}$$

$$\leq \quad \binom{n}{\ell+k}\binom{\ell+k}{k}(\Pr[B_i \in T_{S'}])^\ell \tag{9}$$

$$\leq \quad \binom{n}{\ell+k}\binom{\ell+k}{k}\left( \frac{2em}{n} \right)^{(\ell m)/(2k)} \tag{10}$$

$$\leq \quad \left( \frac{ne}{k+\ell} \right)^{k+\ell}\left( \frac{(\ell+k)e}{k} \right)^{k}\left( \frac{2em}{n} \right)^{(\ell m)/(2k)} \tag{11}$$

$$\leq \quad \left( \frac{ne}{k} \right)^{2k+\ell}\left( \frac{2em}{n} \right)^{(\ell m)/(2k)}, \tag{12}$$

and we can make this final quantity less than 1 by choosing $m = ck \log n$ for an appropriately large constant $c$, using our assumptions that $\ell = \Omega(k)$ and $k = o(n/(\log n))$. $\qquad \square$