[Reviews · NeurIPS 2019]

Reviewer 1



Quality: - Overall the paper is high-quality. - Some lingering questions I have: -- In the experiments, it appears that the algorithms have oracle knowledge of the exact sparsity k. Do these methods work without oracle knowledge of k? It appears that this oracle knowledge of k is used directly in setting measurement parameters, such as the entries of M1, which is unrealistic since the true value of k is rarely known a priori. -- How does this approach perform in the presence of noise? Can the authors run more experiments with noise on the 1-bit measurements? -- Why do the authors only compare against “all Gaussian” measurements? Surely there must be other methods that take measurements in two-stages. In particular the authors reference the approaches in (Acharya et al. 2017) and (Gopi et al. 2013) several times but do not run experimental comparisons against these methods. Originality: - The idea of taking measurements in two stages in compressed sensing (to recover the support, then approximate the signal values on this support) is not new. However, the idea of recovering a superset of the support rather than the exact support appears to be novel. - The connections to Group Testing are interesting and appear to be novel as far as I know. Significance: - Tighter upper bounds than existing results, along with explicit measurement construction methods, are provided for the important settings of non-negative signals (such as images) and binary signals. Even if the idea of taking compressed sensing measurements in two stages is not new, these improvements are important for pushing forward 1bCS research. Clarity: - The paper is mostly clear. - The authors did a good job of juxtaposing their results with previous open questions in the literature - Table 1 clearly highlights the novel contributions -------------------------- Update after rebuttal. The authors comments were satisfactory, as my comments were mostly suggestions for improvement. They didn't mention the points of unclarity in their writing, but I do hope that they will make appropriate corrections on revision.

Reviewer 2



Overall an interesting paper. However, there are several issues that I would recommend the authors address. Most of the issues revolve around the robustness of their approach. In particular, there proposed approach critically depends on the support superset recovery, in order to proceed to the next step of signal recovery. Misidentification of the support will result to incorrect recovery. Furthermore, the larger the identified support the more measurements will be required to fully reconstruct the signal based on a larger support. It is not clear how group testing behaves in the presence of measurement noise. Furthermore, quite often practical signals are not exactly sparse. i.e., the support might be the whole signal even though the signal is sparse because the coefficients decay very fast. Thus, some are approximately, but not exactly, sparse (often known as compressible). Images are such an example. Is group testing robust to this? Will the support returned represent the largest coefficients? If the group testing fails, so will the rest of the process. Another issue is that, it seems, the process requires knowledge of k, to design the group testing. This might often not be available, (admittedly, also a problem with some conventional algorithms, such as the BIHT). However, conventional algorithms seem to be more robust to lack of knowledge of this parameter than group testing ones. Is that an issue? In addition, the experiments are not as convincing as I would expect. The improvements are modest, if any, to justify the extra complications of the two-step process. Also, I would recommend the authors include a line (i.e., a third experiment) in which the superset method matrix is used whole with BIHT (i.e., without the support recovery step first). It might be that the matrix they designed is actually better than a fully Gaussian matrix, even when used with conventional methods. ===== I've seen and taken into account the author's response, and it does not change my score.

Reviewer 3



This is a generally well-written paper with a nice set of results. A potential weakness is that some of the main results come across as rather simple combinations of existing ideas/results, but on the other hand the simplicity can also be viewed as a strength. I don’t find the Experiments section essential, and would have been equally happy to have this as a purely theory paper. But the experiments don’t hurt either. My remaining comments are mostly quite minor – I will put a * next to those where I prefer a response, and any other responses are optional: [*] p2: Please justify the claim “optimal number of measurements” - in particular highlighting the k*log(n/k) + 1/eps lower bound from [1] and adding it to Table 1. As far as I know, it is an open problem as to whether the k^{3/2} term is unavoidable in the binary setting - is this correct? (If not, again please include a citation and add to Table 1) - p2: epsilon is used without being defined (and also the phrase “approximate recovery”) - p4: Avoid the uses of the word “necessary”, since these are only sufficient conditions. Similarly, in Lemma 3 the statement “provided that” is strictly speaking incorrect (e.g., m = 0 satisfies the statement given). - The proof of Lemma 1 is a bit confusing, and could be re-worded. - p6: The terminology “rate”, “relative distance”, and notation H_q(delta) should not be assumed familiar for a NeurIPS audience. - I think the proof of Theorem 10 should be revised. Please give brief explanations for the steps (e.g., the step after qd = (…) follows by re-arranging the choice of n, etc.) [*] In fact, I couldn’t quite follow the last step – substituting q=O(k/alpha) is clear, but why is the denominator also proportional to alpha/k? (A definition of H_q would have helped here) - Lemma 12: Please emphasize that m is known but x is not – this seems crucial. - For the authors’ interest, there are some more recent refined bounds on the “for-each” setting such as “Limits on Support Recovery with Probabilistic Models: An Information-Theoretic Framework” and “Sparse Classification: A Scalable Discrete Optimization Perspective”, though since the emphasis of this paper is on the “for-all” setting, mentioning these is not essential. Very minor comments: - No need for capitalization in “Group Testing” - Give a citation when group testing first mentioned on p3 - p3: Remove the word “typical” from “the typical group testing measurement”, I think it only increases ambiguity/confusion. - Lemma 1: Is “\cdot” an inner product? Please make it clear. Also, should it be mx or m^T x inside the sign(.)? - Theorem 8: Rename delta to beta to avoid inconsistency with delta in Theorem 7. Also, is a “for all d” statement needed? - Just before Section 4.2, perhaps re-iterate that the constructions for [1] were non-explicit (hence highlighting the value of Theorem 10). - p7: “very low probability” -> “zero probability” - p7: “This connection was known previously” -> Add citation - p10: Please give a citation for Pr[sign = sign] = (… cos^-1 formula …). === POST-REVIEW COMMENTS: The responses were all as I had assumed them to be when stating my previous score, so naturally my score is unchanged. Overall a good paper, with the main limitation probably being the level of novelty.

[Author Response · NeurIPS 2019]

First off, thanks to all our reviewers for taking the time to look over our submission and provide thoughtful feedback.
Individual responses are below.

*Response to Reviewer 1:*

**Oracle Knowledge of** $k$**:** In our superset method, you are correct that oracle knowledge of $k$ is assumed both in
determining the Bernoulli probability of the matrix entries and in determining the number of rows of the matrix.
However, in the absence of exact knowledge of $k$, we can instead treat these quantities as hyperparameters to be
optimized; for the Bernoulli probability, we give some experimental evidence in Appendix B that this parameter is not
overly sensitive to tuning, and likely any reasonable estimate of $k$ could be used in place of the exact value. Similar
assumptions have been made in previous work such as the BIHT algorithm, but finding new algorithms that work
independently of the value of $k$ is an interesting problem.

**Noise:** We can extend our methods to handle the case of noisy measurements by using "error-correcting list disjunct
matrices" (see "Efficiently decodable error-correcting list disjunct matrices and applications" Ngo, Porat and Rudra
2011) in place of standard list disjunct matrices in order to recover the superset of the support, at the cost of slightly
more measurements. Since the measurement outcomes are only bits, the *bit-flip* error is the best noise model in this
setting, which can be taken care of by the error-correcting list disjunct matrices. Running some further experiments
with these matrices and measurement noise would be a good addition.

**Comparison with Other Methods:** The main reason we have not attempted to give experimental comparison with
these methods is that they are designed specifically for the universal setting. In the experimental setting we consider,
the measurement matrices have far fewer measurements than what is required for them to be universal; our superset
method gives a fairly smooth degradation of performance even when there is a small measurement budget, but this is
not the case for other methods. The method of Acharya et al., for instance, with less measurements than what is needed
for universality, will have false negatives in the support recovery phase, which results in large error in the second stage.
We could provide this comparison experimentally if it would be beneficial, though.

*Response to Reviewer 2:*

**Robustness:** While badly misidentifying the superset would indeed lead to poor performance in the second stage, this
is only a serious issue if we have false negatives (i.e. we miss coordinates that were in fact in the support). The way our
decoding works, this will never happen, there will be only false positives. So in the worst case, the superset just ends up
being slightly larger, and somewhat more work has to be done in the second stage. For robustness to measurement
noise, see the comments to Reviewer 1 ("Noise"). Handling approximately sparse signals is also an interesting direction
– we believe this could be dealt with by slightly modifying the quantization scheme, so that measurements of very
small magnitude are treated as having zero magnitude. You are correct that as is, our method will not be robust to
approximately sparse signals.

**Knowledge of** $k$**:** Please see comments to Reviewer 1 ("Oracle Knowledge of $k$"). A reasonable estimate of $k$ will
suffice in place of exact knowledge.

**Experiments:** The idea of using the superset matrix directly with BIHT is a very interesting one, that we will consider
in the future version (we believe getting a theoretical result may be challenging). Perhaps using some other matrices
such as an all 0/1 matrix with Bernoulli entries would be worth checking as well.

*Response to Reviewer 3:*

**Wording of "optimal number of measurements":** You are correct, in both the binary and nonnegative cases it is open
whether or not the $k^{3/2}$ term is necessary, so it is unclear whether the number of measurements we obtain is optimal;
thanks for the catch. The lower bound of $k \log(n/k) + (k/\epsilon)$ from Acharya et al. does not obviously apply in either of
these restricted settings, but it is included in Table 1 for the case of real vectors.

**Proof of Theorem 10:** The last step follows from substituting $q = O(k/\alpha)$ as you mention, and also that $1/(1 -$
$H_q((k - \alpha)/k)) = \Theta((\alpha/k) \log(\alpha/k))$, which can be seen by expanding out the terms (using the definition $H_q(p) =$
$-p \log_q(p) - (1 - p) \log_q(1 - p) + p \log_q(q - 1)$). We will include this definition and the details of this step for the
future.

**Novelty:** We believe the primary novelty in the submission is in the use of superset recovery in the 1bCS setting, and
also in developing the relationships between some of the tools and methods used in group testing and those used in
1bCS.

Thanks also for the further references.

[Meta-Review · NeurIPS 2019]

The paper introduces novel techniques for "for-all" one-bit compressed sensing. It improves on the existing results, for instance by showing approximate recovery with k*log(n/k) + 1/eps measurements for eps-recovery, and by extending the O(k^2 log n) support recovery guarantee to explicit constructions. While the novelty is very limited, it is a compelling result for the compressive sensing community.